Genome-wide identification of Apetala2 gene family in Hypericum perforatum L and expression profiles in response to different abiotic and hormonal treatments

Li Yonghui 1 liyonghui@lynu.edu.cn
Chen Yao 1
Yi Ruyi 1
Yu Xueting 1
Guo Xiangmeng 1
YiLin Fan 2
Zhou Xiao-Jun 1 zhouxiaojun@lynu.edu.cn
Ya Huiyuan 3
Yu Xiangli 1
1 School of Life Sciences, Luoyang Normal University , Luoyang, Henan , China
2 Technical Center of zhengzhou Customs Distric , Zhengzhou, Henan , China
3 School of Food and Drug, Luoyang Normal University , Luoyang, Henan , China
Uversky Vladimir
Electronic publication date: 2023 Aug 28
Publication date: 2023
Volume: 11
Electronic Location ID: e15883
Received 2023 May 17; Accepted 2023 Jul 20
Copyright: © 2023 Li et al.
Copyright year: 2023
Copyright holder: Li et al.
License: This is an open access article distributed under the terms of the Creative Commons Attribution License, which permits unrestricted use, distribution, reproduction and adaptation in any medium and for any purpose provided that it is properly attributed. For attribution, the original author(s), title, publication source (PeerJ) and either DOI or URL of the article must be cited.
License URL: https://creativecommons.org/licenses/by/4.0/

Keywords: Hypericum perforatum, AP2 gene family, Abiotic stress, Hormone treatments, Cis-acting elements, Expression pattern analysis

Funding: National Natural Science Foundation of China 31870697 This work was supported by Project of General Project of National Natural Science Foundation of China (Grant Number: 31870697). The funders had no role in study design, data collection and analysis, decision to publish, or preparation of the manuscript.

==============================
The Apetala2 (AP2) gene family of transcription factors (TFs) play important functions in plant development, hormonal response, and abiotic stress. To reveal the biological functions and the expression profiles of AP2 genes in Hypericum perforatum, genome-wide identification of HpAP2 family members was conducted.

Methods

We identified 21 AP2 TFs in H. perforatum using bioinformatic methods; their physical and chemical properties, gene structures, conserved motifs, evolutionary relationships, cis-acting elements, and expression patterns were investigated.

Results

We found that based on the structural characteristics and evolutionary relationships, the HpAP2 gene family can be divided into three subclasses: euANT, baselANT, and euAP2. A canonical HpAP2 TF shared a conserved protein structure, while a unique motif 6 was found in HpAP2_1, HpAP2_4, and HpAP2_5 from the euANT subgroup, indicating potential biological and regulatory functions of these genes. Furthermore, a total of 59 cis-acting elements were identified, most of which were associated with growth, development, and resistance to stress in plants. Transcriptomics data showed that 57.14% of the genes in the AP2 family were differentially expressed in four organs. For example, HpAP2_18 was specifically expressed in roots and stems, whereas HpAP2_17 and HpAP2_11 were specifically expressed in leaves and flowers, respectively. HpAP2_5, HpAP2_11, and HpAP2_18 showed tissue-specific expression patterns and responded positively to hormones and abiotic stresses.

Conclusion

These results demonstrated that the HpAP2 family genes are involved in diverse developmental processes and generate responses to abiotic stress conditions in H. perforatum. This article, for the first time, reports the identification and expression profiles of the AP2 family genes in H. perforatum, laying the foundation for future functional studies with these genes.

Introduction

Hypericum perforatum (common name: St. John’s wort) is a perennial herb in the Hypericaceae family and contains multiple medicinal ingredients (Galeotti, 2017). H. perforatum is a species native to Europe, the Middle East, and North Africa, and it has now adapted to different ecological conditions and climates (Sarrou et al., 2018). The Hypericum genus contains nearly 500 species of representative plants. Among them, H. perforatum is the best-known (Crockett & Robson, 2011). The secondary metabolites in H. perforatum are associated with valuable pharmacological activities against several conditions, such as moderate depression (anxiety and depression) (Galeotti, 2017; Ng, Venkatanarayanan & Ho, 2017; Barnesa, Arnasonb & Roufogalisc, 2019), viral diseases (Chen et al., 2019), oxidative stress (Kováčik et al., 2020), bacterial diseases (Radulović et al., 2018) and cancer (De-Morais et al., 2020); due to this, these metabolites have gained the attention of many researchers. The biological extract of H. perforatum can be mainly divided into three parts, including naphthodianthrones, phloroglucinol, and flavonoids (Nigutová et al., 2019), and the main medicinally active ingredient is naphthodianthrones hypericin, which has been reported to damage the surface structure of the viruses (Wang, Suolang & Ma, 2005).

Plants can regulate their growth and development in response to different abiotic factors, such as drought, cold, and salt, by regulating the expression of a number of genes with specific functions (Wessler, 2005; Kong et al., 2018). Transcription factors (TFs), which play key roles in the gene regulation pathway by directly turning on or shutting off target genes, are involved in the crosstalk between different signaling pathways (Feng et al., 2015; Qin et al., 2017). Multiple TF families with different functional domains are involved in the biosynthetic pathway of major secondary metabolites, such as Apetala2 (AP2)/Ethylene responsive factor (ERF), basic helix-loop-helix (bHLH), WRKY, MYB, and NAC. AP2/ERF constitutes a large TF family in plants, and their members contain at least one AP2 domain (Ramegowda et al., 2014). According to the number and sequence similarity among different AP2 domains, the AP2/ERF superfamily is divided into five families: AP2, dehydration responsive element binding protein (DREB), ERF, related to ABI3/VP1 (Rav), and Soloist (Licausi, Ohme-Takagi & Perata, 2013). TFs in the AP2 subfamily contain two highly conserved AP2 domains, AP2-R1 and AP2-R2, which regulate plant growth and development (Houston et al., 2013). Based on insertion sequences of these two AP2 domains, the HpAP2 gene family can be further divided into three subclasses, including eu-AINTEGUMENTA ANT, baselANT, and euAP2 (Zhao et al., 2019; Dipp-Alvarez & Cruz-Ramirez, 2019). The mRNA of the euAP2 subclass has miR172 target sequences in the post-domain region; further, the ANT lineage was characterized by the insertion of 10 or one amino acids in the AP2-R1 or R2, respectively. The main difference between eu and basalANT is that euANT proteins have a long pro-domain and four conserved motifs (Dipp-Alvarez & Cruz-Ramirez, 2019).

Coen & Meyerowitz (1991) proposed the famous “ABC model” of angiosperm flower development. According to this model, the formation and development of flower organs are determined by three functional genes, A, B, and C. Although AP2 transcripts are ubiquitously distributed in developing flowers, the function of AP2 is limited to the formation of the first and second whorls of sepals and petals (Irish, 2017). AP2 TFs bind to the target sequences of GCAC (A/G) n (A/T) TCCC (A/G) or (C/T) and regulate plant development, such as flower and root growth, seed formation, and bud and ovule development (Aukerman & Sakai, 2003; Nole-Wilson, Tranby & Krizek, 2005; Beth, 2009; Scheres & Krizek, 2018). The homologous gene of AtAP2, target of eat (TOE), has been reported to control the flowering time of Arabidopsis thaliana (common mouse-ear cress) by inhibiting the expression of flowering regulating genes (Mathieu et al., 2009; Yant et al., 2010). In wheat, the AP2 gene Q regulates spike and ear development and reduces plant height (Simons et al., 2005; Greenwood et al., 2017). In rice, two AP2 family genes, SNB and OsIDS1, regulate the establishment of the flower meristem and the growth of inflorescence (Lee & An, 2012). Recently, it has been reported that auxin induces the expression of the AP2 gene and coregulates plant organ development with downstream growth regulators. For example, the drought resistance of A. thaliana was significantly enhanced by the overexpression of the ANT gene (Meng et al., 2015; Krizek, 2015). Furthermore, the AP2/ERF superfamily also plays roles in secondary metabolism, especially in the biosynthesis of the main active components in medicinal plants, such as artemisinin, paclitaxel, and lignin (Wuddineh et al., 2015). AP2 genes also affect the responses of plants to abiotic stresses (such as drought, high temperature, and salt) (Ohto et al., 2005; Meng et al., 2015). For example, the ectopic expression of the CAP2 gene in Cicer arietinum improved the salt and drought resistance in transgenic tobacco and yeast (Shukla et al., 2006).

Genome sequences of Arabidopsis, Poplar, Sorghum, Lingbao rhododendron, and several other species are now available (Velasco et al., 2010; Zhou et al., 2022). The identification and function of AP2/ERF TFs have also been extensively studied in many plants, for instance, barley (Zhuang et al., 2011a; Guo et al., 2016), grape (Licausi et al., 2010), rice (Riano-Pachón et al., 2007), apple (Zhuang, Yao & Xiong, 2011), wheat (Zhuang et al., 2011b), durum wheat (Faraji et al., 2020), potato (Massa et al., 2011), castor bean (Xu et al., 2013), corn (Zhuang et al., 2010), melon (Ma et al., 2015), tomato (Yang et al., 2021), cucumbers (Hu & Liu, 2011), dendrobium officinale (Zeng et al., 2021), soybean (Wang et al., 2022; Jiang et al., 2020), Salvia miltiorrhiza (Ji et al., 2016), Medicago sativa (Jin et al., 2019), and Pisum sativum (Jarambasa et al., 2023), leading to a better understanding of the AP2 TFs. However, the characteristics of the AP2 gene family and the expression pattern in response to hormone or abiotic treatment have not been explored in H. perforatum. Recently, the genome of H. perforatum has been sequenced (Zhou et al., 2021). This study aimed to characterize the expression profiles of AP2 family genes in H. perforatum using the genome sequence data. Our results revealed the gene structures, evolutionary relationships, cis-elements, conserved domains, expression of relevant genes in four tissues (root, stem, leaf, and flower), and expression profiles of genes in response to abiotic factors and hormone treatments. Further, the stress-responsive genes in the H. perforatum AP2 family were identified. This study may provide a theoretical reference for the molecular mechanism of AP2 family genes in response to multiple signals and develop a theoretical foundation for detailed research on the function of AP2 genes in H. perforatum.

Materials and Methods

Identification and sequence analysis of HpAP2 genes

The predicted gene and protein sequences of 30 A. thaliana AP2 genes were obtained from the A. thaliana information system data repository (TAIR, http://www.arabidopsis.org/). The TBLASTX algorithm (https://blast.ncbi.nlm.nih.gov/Blast.cgi?PROGRAM=tblastx&PAGE_TYPE=BlastSearch&BLAST_SPEC=&LINK_LOC=blasttab&LAST_PAGE=blastx) was used to search and compare these sequences against the genome of H. perforatum available in the NCBI database (PRJNA588586). Moreover, the Hidden Markov Model (HMM) of the AP2 DNA-binding domain (PF00847) was used to compare all HpAP2 gene sequences using the Pfam database (https://pfam.xfam.org/) (Zhou et al., 2021). The sequences of selected AP2 genes were validated using InterPro (http://www.ebi.ac.uk/interpro/) and Pfam. The protein sequences of AP2 from A. thaliana, Indian rice (Oryza sativa), and maize (Zea mays) were downloaded from Plant TFDB (http://planttfdb.gao-lab.org/index.php), and conserved domains were identified using InterPro and Pfam. Subsequently, a phylogenetic tree was constructed using a bootstrap analysis with 1,000 replicates in MEGA 7.0, and the results were used to compare the evolutionary relationships between AP2 genes from different plant species. Finally, using 21 predicted AP2 protein sequences of H. perforatum, a phylogenetic tree was constructed as described above.

Protein structure prediction and Ka/Ks analysis

The amino acid number, subcellular localization, relative molecular mass, isoelectric point (PI), hydrophilicity coefficient, instability coefficient, and liposolubility index of HpAP2 proteins were predicted using ExPASy (http://web.ExPASy.org/compute_pi/) and CELLO (http://cello.life.nctu.edu.tw/). Based on the genomic and corresponding coding sequences (CDS), Gene Structure Display Server (GSDS) program was used to graphically display the exon and intron structures of the AP2 family genes (Hu et al., 2015). The conserved motif sequences in the HpAP2 family proteins were determined using the MEME (https://meme-suite.org/meme/tools/meme), with the forecast range set to 12 motifs. The ratios of non-synonymous substitution rate (Ka) to synonymous substitution rate (Ks) of the HpAP2 amino acid sequences were determined using the KaKs_Calculator 2.0. The Ka/Ks values reflect the evolutionary selection pressure for gene pairs and informs on the AP2 gene’s selection pattern among paralogous genes of H. perforatum.

Cis-acting elements and transcriptomic analysis

Considering the promoter sequence 1,500 bp upstream of the start codon of the HpAP2 gene, the potential cis-acting elements were searched and analyzed using the PlantCARE database (Lescot et al., 2002). RNA-seq data were retrieved from the SRA-NCBI database (https://trace.ncbi.nlm.nih.gov/Traces/sra/) (Zhou et al., 2021), and the serial login numbers of the flower, root, leaf, and stem were SRR8438983, SRR8438986, SRR8438984, and SRR8438985.

Plant materials and stress treatments

The H. perforatum wild seeds (2n = 2x = 16) from the Qinling Mountains in Cheng County, Gansu Province, were placed (after screening and removing impurities) in a 1.5 mL centrifuge tube, sterilized with 10% sodium hypochlorite for 10–15 min, and washed 7–10 times with sterile water. The seeds were germinated on the Murashige and Skoog (MS) solid medium under natural light (16 h light and 8 h dark) at 25 °C. The germinated seedlings were selected for gene expression profiling. For salt and drought treatments, seedlings were placed in sodium chloride (200 mM) or polyethylene glycol (PEG) (20%) solutions, respectively. Seedlings were cultured at 4 °C for low temperature stress. For hormone treatments, 2-month-old aseptic seedlings were placed in MS medium containing 100 µmol/L of Gibberellin (GA) or abscisic acid (ABA). After 0, 1, 3, 6, or 12 h of hormone treatment, the triplicate samples were collected, frozen in liquid nitrogen, and stored at −80 °C.

Quantitative real-time reverse transcription PCR (qRT-PCR)

The total RNA was extracted from H. perforatum seedlings according to the protocol mentioned in the TaKaRa RNA extraction kit (Japan, Dalian), and the RNA concentration was measured using a NanoDrop 2000 ultramicro spectrophotometer. The RNA integrity was detected via 1% agarose gel electrophoresis. The RNA samples with a wavelength ratios of A260/A280 and A260/A230 close to 2.0 were used for cDNA synthesis. Subsequently, the total RNA (1.0 μg) was reverse transcribed using a PrimeScript™ RT Reagent Kit (TaKaRa, Beijing, China) in a 20 μL reaction volume. qRT-PCR (which was conducted in 20 µL reactions), the dissolution curve analysis was carried out under the following conditions: 55 °C for 10 s and 98 °C for 5 s. Table S2 lists the primers used. Each reaction had a negative control group and three biological and technical replicates. Using actin (HpACT2) as the internal reference gene, the relative expression of HpAP2 was calculated using the 2−ΔΔCt method. One-way analysis of variance and mapping were performed using GraphPad Prism 9.3 software.

Results and analysis

Genome-wide identification and sequence characterization of HpAP2 family

A total of 21 HpAP2 gene sequences containing the AP2 domain were identified. All the sequences (named HpAP2_1 to HpAP2_21) were verified and confirmed using InterPro and Pfam. In the ExPASy analysis, no significant difference was observed in the number of amino acids of different AP2 proteins; maximum, minimum, and average lengths were 625 (HpAP2_3), 341 (HpAP2_19), and 497 amino acids, respectively (Table 1). The molecular weight of the 21 predicted AP2 proteins of H. perforatum ranged from 38.74385 kDa (HpAP2_19) to 68.41809 kDa (HpAP2_3), and the average isoelectric point was 6.9, while seven proteins were >7.0 and 14 proteins were <7.0. The amount of HpAP2 family TFs (21) were higher than that of Arabidopsis (14) and slightly lower than that of Indian rice (24) and maize (23).

Table 1 Characteristics of AP2 genes in H. perforatum.

Gene ID	Subcellular
positioning	Amino acid number	Relative molecular weight	Isoelectric point PI	Hydrophilic
coefficient	Instability coefficient	Lipid solubility index	
HpAP2_1	Nucleus	619	68,324.41	6.56	−0.654	49.41	56.19	
HpAP2_2	Nucleus	474	52,166.53	6.16	−0.685	42.33	55.23	
HpAP2_3	Nucleus	625	68,418.09	6.20	−0.741	49.25	54.08	
HpAP2_4	Nucleus	450	49,286.55	7.18	−0.616	52.52	62.73	
HpAP2_5	Nucleus	423	46,741.06	7.53	−0.725	57.06	57.68	
HpAP2_6	Nucleus	409	45,452.68	9.19	−0.628	51.31	63.96	
HpAP2_7	Nucleus	560	60,895.78	6.37	−0.632	46.55	60.36	
HpAP2_8	Nucleus	356	40,357.67	7.11	−0.790	65.58	57.84	
HpAP2_9	Nucleus	548	60,307.10	6.21	−0.674	46.64	54.20	
HpAP2_10	Nucleus	400	45,022.51	9.16	−0.735	63.26	63.47	
HpAP2_11	Nucleus	546	60,504.19	6.27	−0.965	64.14	49.19	
HpAP2_12	Nucleus	356	40,312.63	6.80	−0.769	61.78	58.93	
HpAP2_13	Nucleus	624	68,318.96	6.20	−0.745	48.92	53.85	
HpAP2_14	Nucleus	605	66,580.02	6.00	−0.762	51.77	53.87	
HpAP2_15	Nucleus	534	58,870.49	6.07	−0.729	46.95	57.96	
HpAP2_16	Nucleus	596	65,481.74	6.00	−0.777	52.44	54.51	
HpAP2_17	Nucleus	440	48,738.41	8.13	−0.573	57.17	62.34	
HpAP2_18	Nucleus	594	65,411.33	6.96	−0.621	46.97	63.77	
HpAP2_19	Nucleus	341	38,743.85	8.54	−0.997	60.06	54.13	
HpAP2_20	Nucleus	375	42,321.71	6.77	−0.885	57.58	53.68	
HpAP2_21	Nucleus	560	60,925.81	6.37	−0.633	45.76	60.18	

Phylogenetic relationships of different AP2 families

A total of 82 AP2 TFs were selected from four different species, including H. perforatum (21), A. thaliana (18), Indian rice (24), and maize (23), and the corresponding phylogenetic trees were constructed (Fig. 1). A. thaliana was selected because it is a model dicot plant and has been extensively studies. Rice, which is a model monocot plant, and maize were selected because they both are important food crops. Three subfamilies of AP2 TFs were identified, and the size order of the three subfamilies was euANT, baselANT, and euAP2, which contained 11, six, or four HpAP2 genes, respectively. Furthermore, most of the AP2 TFs of A. thaliana and H. perforatum were clustered in the same branch, indicating a closer relation between them compared to Indian rice and maize. AP2 family members of Indian rice and maize were distributed in the same branch, indicating that the genes from the two monocot species had a higher homology.

Figure 1 Phylogenic cluster of AP2 families in H. perforatum (21), A. thaliana (18), Oryza sativa (24), and Zea mays (23).

Three AP2 subclasses (the euANT, baselANT, and euAP2) are shown inside the outer ring.

Gene structure and conserved domains of HpAP2

A new phylogenetic map was generated using the HpAP2 protein sequence (Fig. 2). Further, the gene structure diagram (Fig. 3) revealed that the minimum and maximum number of introns were five and nine, respectively, and the gene structures were similar among homologous genes. The intron and exon lengths of HpAP2 in the same subgroup were different, but the structure was highly conserved (for instance, the euANT subgroup reached 87.5%). Further, 12 conserved motifs were identified in HpAP2 TFs, which were named as motifs 1–12 (Fig. 4). All the AP2 members were found to contain motifs 1–4, suggesting a high conservation of these motifs. Motifs 5, 9, and 10 were also common in HpAP2 members. These results demonstrated identical motif composition of evolutionarily related genes, indicating that AP2 TFs in the same subgroup had similar effects.

Figure 2 Twenty-one AP2 proteins of H. perforatum clustered into a phylogenetic map (left) and multiple sequence alignment of euAP2 of H. perforatum (right).

Figure 3 The phylogenetic tree (left) prepared using sequences of 21 AP2 proteins from H. perforatum. The genetic map is shown (right).

Orange box, black line, and blue box represent CDS, introns, and upstream region, respectively.

Figure 4 Phylogenetic relationships and composition of conserved motifs in AP2s of H. perforatum.

(A) The motif patterns of 21 HpAP2 proteins. Each motif is shown by the box in different colors. (B) Sequence logos of each motif.

Analysis of cis-acting elements

A total of 59 cis-elements were detected (Fig. 5). Most elements were related to stress responses, including hormone response elements (such as MeJA, ABA, SA, GA, and IAA), and abiotic factors (such as cold, damage, hot, and light). Further, ABRE (CGTCA-motif) and methyl jasmonate (TGACG-motif) response elements were more than other hormone regulatory elements. A total of ten HPAP2s contained W-box (TTGACC) element that binds to WRKY TFs, which regulate wound, pathogen reactivity, and gene expression. They are involved in wound and pathogenic bacteria-related stress responses and regulate their own activity or cross-talk with other signaling pathways (Chi et al., 2013; Jiang et al., 2016). In HPAP2 family, a large number of cis-acting elements were observed in the promoter region. These included G-box, Box4, light response (TCT-motif), cold response element (LTR-motif), and drought-induced element (MYB binding site, MBS) in the promoter regions of genes 14, 19, 9, 8, and 12, respectively.

Figure 5 Cis-acting element prediction of HpAP2 gene family promoters.

The map shows the number (Y-axis) of identified cis-elements in relation to specific conditions/elicitors/processes (X-axis) in the HpAP2 gene family members.

Driving forces of genetic differentiation

According to the cluster dendrogram, there were eight highly homologous relative gene pairs in the HPAP2 gene family (Fig. 2), suggesting that almost 76% of the AP2 genes were duplicated. It indicates that functional diversity and gene family expansion occurred in most genes during evolution. The values of Ks, Ka, and Ka/Ks of the AP2 genes are shown in Table S1. In general, the Ka/Ks ratio of <1 indicated that the gene pair was in a negative selection or purification selection state. Further, Ka/Ks ratio of 1 and >1 indicated neutral and positive selection, respectively (Wang et al., 2018). In this study, except for HpAP2_1/HpAP2_18, HpAP2_2/HpAP2_9, and HpAP2_14/HpAP2_ 16, the Ka/Ks ratio of the remaining five gene pairs was <1, indicating that most of the HpAP2 genes had undergone purification under selection pressures. The Ka/Ks ratios of the three pairs HpAP2_1/HpAP2_18, HpAP2_2/HpAP2_9, and HpAP2_14/ HpAP2_16 were greater than 1, with values of 1.0757, 3.0730, and 1.0707, respectively, and indicated a strong positive selection.

Transcriptomics results

The transcriptomics data of the root, stem, leaf, and flower of H. perforatum were used to explore the expression profiles of HpAP2 genes in these tissues (Fig. 6). The RNA-seq results were retrieved from the SRA-NCBI database. We conducted hierarchical clustering based on the expression data and generated the heat map to visualize the expressions of HpAP2 genes (Fig. 6). The transcripts of two HpAP2s (HpAP2_7 and HpAP2_20) were not detected in any of the four types of tissues, indicating that these genes might be pseudogenes. A total of six AP2 genes with relatively higher expression levels were detected in tissues. The six genes were HpAP2_1, HpAP2_4, HpAP2_5, HpAP2_11, HpAP2_17, and HpAP2_18. In addition, few AP2 genes were predominantly expressed in one or more tissues. For example, HpAP2_17 was highly expressed in all tissues. HpAP2_1, HpAP2_4, HpAP2_9, HpAP2_11, and HpAP2_12 were highly expressed in flowers, whereas HpAP2_1, HpAP2_5, and HpAP2_18 were highly expressed in stems. Finally, HpAP2_4 was highly expressed in leaves.

Figure 6 Tissue-specific expression analysis of HpAP2 genes.

The color code shown on the top of the figure represents different log10 values.

Analysis of expression patterns under abiotic stress

A total of five stress resistance HpAP2 subfamily genes were selected for expression pattern analysis under stress conditions (Fig. 1). After treated with different abiotic stresses (drought, salt, and cold) or exogenous hormones (ABA and GA) for 0, 1, 3, 6, and 12 h, the expression levels of the five HpAP2 genes were determined. As shown in Fig. 7, HpAP2_12 was not responsive to any of the treatments, while HpAP2_5, HpAP2_11, and HpAP2_18 were differentially expressed in H. perforatum. The expression level of HPAP2_17 was upregulated at low temperature (4 °C) and in presence of PEG or GA, while down-regulated when treated with NaCl or ABA. Under cold stress, HpAP2_11 was up-regulated approximately 21-fold at 3 h, whereas HpAP2_18 was up-regulated approximately 35-, 17-, or 57-fold at 1, 3, or 12 h, respectively. When treated with NaCl, HpAP2_18 was up-regulated more than 100-folds and up to 214-folds at 6 h, representing the strongest induction. Four HpAP2 genes (HpAP2_5, HpAP2_11, HpAP2_17, and HpAP2_18) were upregulated by PEG. Under ABA treatment, HPAP2_5 expression increased by 16-fold after treatment for 12 h. Under GA treatment, the expression of HpAP2_18 was upregulated 25-fold at 6 or 12 h, and 41-fold at 3 h. The major inducible genes in response to cold, NaCl, drought, ABA, and GA were HpAP2_18, HpAP2_18, HPAP2_17, HpAP2_5, and HpAP2_18, respectively.

Figure 7 Expression levels of HpAP2 genes under different treatments.

(A) 4 °C, (B) NaCl, (C) PEG, (D) Abscisic acid (ABA) and (E) Gibberellin (GA). X-axis shows process-time and Y-axis shows the value of relative expression which is conversed by Log2 (Fold change). Statistical analysis was performed using an one-way analysis of variance (*P < 0.05; **P < 0.01; ***P < 0.001).

Discussion

Genome-wide analysis is an approach that is preferably used to characterize the functional genes to better understand the species evolution (Zhang et al., 2015). Although the AP2/ERF TFs have been extensively studied in many species, the AP2 family in H. perforatum remains unexplored. In this study, genomic studies were conducted on AP2 family in H. perforatum and 21 HpAP2 family members were identified. Phylogenetic analysis showed that these genes could be divided into three subfamilies: euANT, baselANT, and euAP2 (Zhao et al., 2019). Identical genetic structures, conserved domains, and phylogenetic analysis of AP2 proteins in the same branch strongly supported the accuracy of AP2 family classification (Fig. 1). In addition, the prediction of subcellular localization indicated that all HpAP2 family proteins were localized in the nucleus (Table 1), which is in line with the transcription functions (Zhou et al., 2020).

Gene structure analysis showed that all the coding sequences of HpAP2 genes were separated by introns. HpAP2 genes contained a minimum and maximum of five and ten introns, respectively. Further, the HpAP2 genes in euANT, basalANT, and euAP2 subgroups had 8–9, 6–7, and 8–10 exons, respectively. However, there were few exceptions. For example, HpAP2_6 lacked two exons compared with other genes in the same subgroup, which may be caused by the deletion or insertion of introns during the formation of the AP2 gene family. Therefore, in the process of evolution, genes might form different exon and intron structures, thus exerting their functions (Rogozin et al., 2005; Wang et al., 2016). The existence of conserved motifs in TFs is important for biological functions (Sakuma et al., 2002). Motif 2–3 (none in HpAP2_6) and 4-1 constituted two AP2/ERF domains in AP2 TFs, which can be utilized as the main feature to identify HpAP2 family. Interestingly, motif 7 was specific for the euANT, basalANT, and euAP2 subgroups. In the euAP2 subgroup, motif 2 was closely related to motif 3. However, in the euANT and basalANT subgroups, a short connection sequence was identified between motif 2 and motif 3; a part of this sequence was found to be made of motif 7. It was also annotated as part of a 10-amino acid insertion sequence in the multiple alignments and was reported to be the major difference between the euANT, basalANT, and euAP2 groups (Kim et al., 2006; Horstman et al., 2014). miR172 is known to regulate euAP2 TFs in rice, Arabidopsis, and Brassica napus through transcriptional cleavage and translation. Previous reports showed that miR172 might regulate euAP2 genes through transcript cutting and translation in rice, Brassica napus and A. thaliana (Schmid et al., 2003; Tang, Li & Chen, 2007; Wang et al., 2019). In this study, all four euAP2 genes contained miRNA recognition element (MRE) domain of miR172, as per the target prediction (Wang et al., 2019) (Fig. 2). In addition, the binding sites of miRNA172 were also found in the 3′ coding region of the AP2 genes and few representative euAP2 proteins in rice and barley (Tang, Li & Chen, 2007; Gilhumanes et al., 2009), indicating that the complementary sites of miR172 in euAP2 genes were conserved in plants. Notably, motifs 2 + 7, 9, and 11 covered all structures of the euANT1 to euANT3, and these motifs were reported to be the dominant features of the euANT class (Fig. 4) (Kim et al., 2006). Furthermore, we found that HpAP2_7, HpAP2_15, and HpAP2_21 homologous genes clustered together with AthAIL3 (AT3G20840) and AthAIL4 (AT1G51180) in euANT group. They all exclusively encoded a motif 6 (Fig. 4). Previous studies have shown that TFs sharing unique motifs in a cluster may have similar efficacy. Therefore, the research of this unique motif 6 may reveal the neofunctionalization of the ANT genes. Our results suggested that during the formation and evolution of the genome of H. perforatum, the HpAP2 family may have undergone genetic differentiation to fulfil different biological functions.

AP2 TFs play important roles in controlling the growth and development of plants through coping with hormone signals and environmental stresses (Meng et al., 2015; Krizek, 2015; Scheres & Krizek, 2018; Jiang et al., 2019). In this study, most HpAP2 genes had multiple cis-acting factors related to abiotic response, showing that HpAP2 genes had potential roles in improving the stress resistance or tolerance of plants (Sakuma et al., 2002). It was also found that the HpAP2 promoter contained a large number of cis-acting elements that respond to various hormonal stimuli, such as methyl jasmonate, ABA, and GA. A total of 12 (57.14%) genes contained a MBS element in the promoter region which responds to drought stress. We showed that HpAP2_17, HpAP2_18 were highly responsive to drought stress (Fig. 7). Furthermore, we found that most HpAP2 genes were involved in hormonal and abiotic stress responses. The Ka/Ks values determined in this study indicated possible evolutionary selection processes for different genes, and hence, the classification of genes based on the Ka/Ks ratio paves the way for future functional analysis (Zhou et al., 2020).

Expression patterns showed that the majority AP2 family genes (57.14%) were extensively and differentially expressed (Fig. 6). For example, in euAP2 subgroup, HpAP2_4, HpAP2_6, HPAP2_11, and HpAP2_17 were highly expressed in roots and flowers, indicating a potential role in root and flower development (Zhao et al., 2019). Studies have shown that the genes of ant, ant-like5, ail6, and ail7 in the euANT subgroup of Arabidopsis regulate floral growth and ovule development (Elliott et al., 1996; Beth, 2009; Krizek, 2011). In this study, the HpAP2_1 of euANT subgroup clustered with ANT (AT1G72570), while HpAP2_2 and HpAP2_9 clustered with AIL6 (AT5G10510) and AIL7 (AT5G65510), respectively (Fig. 1). Further, HpAP2_1 and HpAP2_9 were highly expressed in flowers, while HpAP2_1 and HpAP2_2 were highly expressed in stems and roots, indicating their roles in the development of these organs in H. perforatum. Meanwhile, it was speculated that the different expression patterns between HpAP2_2 and its homologous genes may be due to the sub-functionalization or new functionalization during the evolution of H. perforatum. Our results also showed that several AP2 genes preferentially expresses in some tissues. For example, HpAP2_18 had the highest transcript accumulation in roots and stems compared with other genes, while HpAP2_17 and HpAP2_11 had the highest transcript accumulation in leaves and flowers, respectively. In A. thaliana, the AtAP2 homologous gene TARGETS OF EAT (TOE) participates in the regulatory mechanism of flowering time by inhibiting the flowering genes (Mathieu et al., 2009). Figure 1 shows that HpAP2_17 and AtTOE1 (AT2G28550), HPAP2_11, and AtTOE3 (AT5G67180) could be clustered together. It was speculated that HpAP2_11 and HpAP2_17 may also be involved in the regulation of flowering time of H. perforatum. These results will be helpful to further understand the function of HpAP2_17 and HpAP2_11.

The dry, aboveground parts of H. perforatum have been used in traditional Chinese medicine for the treatment of various conditions, such as depression and cancer. The main active components of H. perforatum are naphthodianthrones, phloro glucinols, flavonoids, xanthon, alkaloids, volatile oil, and many others (Sarrou et al., 2018). Since H. perforatum has many medicinal properties, understanding the molecular structure of genes responsible for the growth and development of this plant is crucial. The overexpression of LaAP2L1, which is a HpAP2 gene homolog in Larix, was found to significantly increase the organ size, biomass, and seed yield in transgenic A. Thaliana plant (Li et al., 2013). Furthermore, the AP2/ERF superfamily has been shown to be involved in regulating the biosynthesis of secondary metabolites in plants, such as terpenes, flavonoids, and alkaloids (Zhou & Memelink, 2016). In H.perforatum, the expression of HpAP2 genes is possibly related to the biomass production and yield, allowing the accumulation of ingredients of medicinal importance. This study showed that the HpAP2 genes were induced in response to abiotic stresses and plant hormones; these findings will be helpful for further studies to understand the role of HpAP2 family genes in the environmental adaptation and hormone-regulated organ development of H. perforatum.

Plants have multiple pathways for sensing, recognizing, and responding to various internal and external signals, and hence, survive under ever-changing external conditions (Zhou et al., 2020). Figure 7 shows that the expression levels of five HpAP2 genes were different under different treatments. According to these results, HpAP2_18 may be involved in responses to abiotic stresses in roots and stems. Similar to AtTOE3, HpAP2_11 was found to preferentially express in flower tissues, and it was also induced by low temperature, NaCl, PEG, ABA, and GA treatments, suggesting that HpAP2_11 may participate in stress responses, specifically in flower tissues. Moreover, HpAP2_5 was significantly expressed in all the four tissues and also induced by low temperature, NaCl, PEG, and ABA treatments. In conclusion, HpAP2_5, HpAP2_11, and HpAP2_18 showed tissue-specific expression, and may play significant roles in generating responses to hormones and environmental abiotic factors in H. perforatum.

Conclusions

The bioinformatics and transcriptomics analyses performed in this study identified, for the first time, 21 key genes of AP2 family in H. perforatum. Further phylogenetic analysis showed that the HpAP2 family can be divided into three subgroups: euANT, baselANT, and euAP2. The members in the same subgroup showed similar gene structures and conserved motifs. Most HpAP2 TFs shared four conserved motifs. According to the expression patterns, we found that HpAP2_5, HpAP2_11, and HpAP2_18 were tissue-specific and responded positively to hormonal stimuli or abiotic stresses. This study laid a foundation for the classification and functional analysis of the AP2 genes in H. perforatum. Further, the study provides valuable clues for developing strategies for enhancing the stress tolerance of H. perforatum.

Ackowledgments

The authors would like to thank all the reviewers who participated in the review, Special thanks to Professor Wang Zhezhi and Dr. Zhou Wen of Northwest China National Engineering Laboratory for Resources Exploitation of Endangered Chinese Medicinal Materials for their guidance and assistance, as well as MJ Editor for providing English editing services during the preparation of this manuscript.

Supplemental Information

Supplemental Information 1 Ks, Ka, and Ka/Ks ratios of the eight gene pairs. Primers used for real-time fluorescent quantitative PCR of stress response genes.

Click here for additional data file.

Supplemental Information 2 Sequences related to HpAP2 bioinformatics analysis.

Click here for additional data file.

Supplemental Information 3 qPCR experiment results.

Click here for additional data file.

Supplemental Information 4 KAKS results.

Click here for additional data file.

Additional Information and Declarations

Competing Interests

Author Contributions

Data Availability

The authors declare that they have no competing interests.

Yonghui Li conceived and designed the experiments, performed the experiments, analyzed the data, prepared figures and/or tables, authored or reviewed drafts of the article, and approved the final draft.

Yao Chen conceived and designed the experiments, analyzed the data, prepared figures and/or tables, authored or reviewed drafts of the article, and approved the final draft.

Ruyi Yi performed the experiments, prepared figures and/or tables, and approved the final draft.

Xueting Yu conceived and designed the experiments, performed the experiments, analyzed the data, prepared figures and/or tables, and approved the final draft.

Xiangmeng Guo performed the experiments, authored or reviewed drafts of the article, and approved the final draft.

Fan YiLin conceived and designed the experiments, authored or reviewed drafts of the article, and approved the final draft.

Xiao-Jun Zhou conceived and designed the experiments, prepared figures and/or tables, authored or reviewed drafts of the article, and approved the final draft.

Huiyuan Ya conceived and designed the experiments, prepared figures and/or tables, authored or reviewed drafts of the article, and approved the final draft.

Xiangli Yu performed the experiments, analyzed the data, authored or reviewed drafts of the article, and approved the final draft.

The following information was supplied regarding data availability:

The raw measurements are available in the Supplemental Files.

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
