# Peer review of "Genome-wide identification of Apetala2 gene family in Hypericum perforatum L and expression profiles in response to different abiotic and hormonal treatments"

_PeerJ, doi:10.7717/peerj.15883_

## Round 0.1 · original submission · Minor Revisions

Please address issues pointed out by reviewers and amend the manuscript accordingly.

Reviewer 1 ·

Basic reporting

The manuscript describes a comprehensive analysis of the H. perforatum AP2 subfamily. This study can be used as a foundation for future studies on AP2 TFs in this crop. Therefore, this paper can be accepted for publication with minor revisions as described below.

The following suggestions can be used to improve the language:

‘cis’ needs to be italicized in each instance of cis-acting regulatory elements
Line 40: what are ‘obvious characteristics’?
Line 98: gene name needs italicizing
Line 102: Double check the pfam family you said you used. PF00841 corresponds to the Protamine_P2 family
Line 105: The sequences were downloaded from Plant TFDB, not Plant Ploc
Line 114: ‘wit’?
Line 121: Shouldn’t it be PlantCARE?
Line 163: misspelling of perforatum
Line 174: instead of ‘indicating’ I would use ‘suggesting’.
Line 203; needs rewording
Line 211: Do you mean ‘Figure 7’?
Line 300: check spelling

Experimental design

1. You should mention where the H. perforatum genome sequences were obtained from.

Validity of the findings

no comment

Additional comments

no comment

Reviewer 2 ·

Basic reporting

Reviewer’s Opinion: The article presents some important findings related to the identification and expression profiles of the AP2 family genes in H. perforatum, however, there are some minor issues that the athors must take-care of, for making it suitable for publication. Comments are as follows:
1. Grammatical Issues: The authors need to thoroughly go through the text to correct the following and more (if any) grammatical mistakes in the draft:
• The sentence: “Furthermore, most of the AP2 TFs of A. thaliana and H. perforaturm were clustered in the same branch, indicating a more close relation between them compared to Indian rice and maize.”, “more close” should be replaced by “closer”, as “more-close” is grammatically incorrect.
• In the sentence: “A total of six AP2 genes with a relatively high expression levels were detected in tissues.”, under “Transcriptomics results” (lines 220 – 221), “a relatively high” should be replaced with “relatively higher”, as the authors are talking about multiple AP2 genes.
• In the sentence:” Although the AP2/ERF TFs have been extensive studied in many different species”, the word “extensive” should be replaced by “extensively” (lines 244-245)
• Lines 258-259 “Therefore, in the process of evolution, genes have formed different exon and intron structures, thus transforming their functions (Rogozin et al., 2005; Wang et al., 2016).” Is too affirmative. “genes have formed” should be replaced with “genes might have formed”, as this is only a speculation reflected from the author’s results.
2. Missing references/Peer recognition in the field:
I. The authors have although given a good introduction about the Apetala2 gene family and research done so far, they have not cited various research works on the genome-wide identification of the Ap2/ERF gene family in other plant species:
1. Jin X, Yin X, Ndayambaza B, Zhang Z, Min X, Lin X, Wang Y, Liu W. Genome-wide identification and expression profiling of the ERF gene family in Medicago sativa L. under various abiotic stresses. DNA and Cell Biology. 2019 Sep 1;38(10):1056-68.
2. Jarambasa T, Regon P, Jyoti SY, Gupta D, Panda SK, Tanti B. Genome-wide identification and expression analysis of the Pisum sativum (L.) APETALA2/ethylene-responsive factor (AP2/ERF) gene family reveals functions in drought and cold stresses. Genetica. 2023 Jun 3:1-5.
3. Guo B, Wei Y, Xu R, Lin S, Luan H, Lv C, Zhang X, Song X, Xu R. Genome-wide analysis of APETALA2/ethylene-responsive factor (AP2/ERF) gene family in barley (Hordeum vulgare L.). PLoS One. 2016 Sep 6;11(9):e0161322.
4. Ma Y, Zhang F, Bade R, Daxibater A, Men Z, Hasi A. Genome-wide identification and phylogenetic analysis of the ERF gene family in melon. Journal of Plant Growth Regulation. 2015 Mar;34:66-77.
5. Yang H, Sun Y, Wang H, Zhao T, Xu X, Jiang J, Li J. Genome-wide identification and functional analysis of the ERF2 gene family in response to disease resistance against Stemphylium lycopersici in tomato. BMC plant biology. 2021 Dec;21:1-3.
6. Ji AJ, Luo HM, Xu ZC, Zhang X, Zhu YJ, Liao BS, Yao H, Song JY, Chen SL. Genome‐Wide Identification of the AP2/ERF Gene Family Involved in Active Constituent Biosynthesis in Salvia miltiorrhiza. The plant genome. 2016 Jul;9(2):plantgenome2015-08.
7. Hu L, Liu S. Genome-wide identification and phylogenetic analysis of the ERF gene family in cucumbers. Genetics and Molecular Biology. 2011;34:624-34.
8. Zeng D, Teixeira da Silva JA, Zhang M, Yu Z, Si C, Zhao C, Dai G, He C, Duan J. Genome-wide identification and analysis of the APETALA2 (AP2) transcription factor in Dendrobium officinale. International Journal of Molecular Sciences. 2021 May 14;22(10):5221.
9. Wang H, Ni D, Shen J, Deng S, Xuan H, Wang C, Xu J, Zhou L, Guo N, Zhao J, Xing H. Genome-wide identification of the AP2/ERF gene family and functional analysis of GmAP2/ERF144 for drought tolerance in soybean. Frontiers in Plant Science. 2022;13.
10. Jiang W, Zhang X, Song X, Yang J, Pang Y. Genome-wide identification and characterization of APETALA2/ethylene-responsive element binding factor superfamily genes in soybean seed development. Frontiers in Plant Science. 2020 Sep 4;11:566647.
11. Faraji S, Filiz E, Kazemitabar SK, Vannozzi A, Palumbo F, Barcaccia G, Heidari P. The AP2/ERF gene family in Triticum durum: genome-wide identification and expression analysis under drought and salinity stresses. Genes. 2020 Dec 7;11(12):1464.

And many more…. The reviewer suggests that the authors must thoroughly check for such citations which are missing from the present draft and very briefly acknowledge such research works.

II. Please cite TBLASTX (when mentioned for the first time) in section “Materials and methods”.
III. Similarly, please cite the SRA-NCBI database, when mentioned for the first time under “Transcriptomics results”, under “Results” section.
3. Repeated references:
The following reference has been repeated twice:
Zhao Y, Ma R, Xu D, Bi H, Xia Z, and Peng H. 2019. Genome-Wide identification and analysis of the AP2 transcription factor gene family in wheat (Triticum aestivum L.). Frontiers in Plant Science 10: 1286 DOI: 10.3389/fpls.2019.01286
The reviewer suggests the authors must thoroughly check their draft for such mistakes in the citations.

Experimental design

Novelty of research: The present work is novel, as this is the first time that a genome-wide analysis of Hypericum perforatum L. has been undertaken. Since the plant species has a wide range of medicinal values, this work will pave the way for further research and development of the HpAP2 family genes.
The authors have although well identified the benefits of the plant species, they must also mention how their study contributes to the medicinal or other research of Hypericum perforatum L. or the HpAP2 family genes. This will strengthen the importance of their publication.

Validity of the findings

1. What is the rationale of the authors to choose only the three plant species like A. thaliana, Indian rice (Oryza sativa), and maize (Zea mays), for comparison? This should be mentioned in the draft.

2. In the “Results” section under “Gene structure and conserved domains of HpAP2” , the authors mention:

“The intron and exon lengths of HpAP2 in the same subgroup were different, but the structure was highly conserved.”
It would be beneficial if the authors can also mention the % conservation to support this claim of “highly conserved”.

Additional comments

I. Please mention the common name of Hypericum perforatum, which is St. John’s Wort. Similarly, please mention the common name for A. thaliana, which is mouse-ear cress.
II. The authors also need to add more medicinal values of the studied species of H. perforatum : like treatment of anxiety and depression, with citing relevant references.
III. In the “Methods and Material” section, under subsection “Protein structure prediction and Ka/Ks analysis”, the authors must very briefly describe the importance of Ka/Ks analysis.

---

## Round 0.2 · accepted · Accept

All concerns of the reviewers are adequately addressed and the revised manuscript is acceptable now.